# Effect of the First Year of COVID-19 Pandemic on Ophthalmological Practice: A Multi-Centre Italian Study with a Focus on Medico-Legal Aspects

Giuseppe Demarinis [1,†], Daniela Mazzuca [2,†], Filippo Tatti [1], Massimiliano Borselli [3], Alessandra Mancini [3], Adriano Carnevali [3], Laura Logozzo [3], Antonio Veraldi [3], Ottavio Stefano [4], Francesca Guarna [2], Vincenzo Scorcia [3], Enrico Peiretti [1,†] and Giuseppe Giannaccare [3,*,†]

1 Department of Surgical Sciences, Eye Clinic, University of Cagliari, Via Ospedale 48, 09124 Cagliari, Italy; demarinis91@gmail.com (G.D.); filippotatti@gmail.com (F.T.); enripei@hotmail.com (E.P.)

2 Department of Surgical and Medical Sciences, University 'Magna Græcia' of Catanzaro, 88100 Catanzaro, Italy; danielamazzuca7@gmail.com (D.M.); francescaguarna@blu.it (F.G.)

3 Department of Ophthalmology, University 'Magna Græcia' of Catanzaro, 88100 Catanzaro, Italy; mborselli93@gmail.com (M.B.); ale.mancini.a@gmail.com (A.M.); adrianocarnevali@unicz.it (A.C.); logozzolau@libero.it (L.L.); a_veraldi@hotmail.it (A.V.); vscorcia@unicz.it (V.S.)

4 Forensic Medicine Specialist, AO of Cosenza, Via Felice Migliori, 87100 Cosenza, Italy; ottavio.stefano@gmail.com

* Correspondence: giuseppe.giannaccare@unicz.it; Tel.: +39-33-1718-6201; Fax: +39-096-1364-7094

† These authors contributed equally to this work.

**Abstract:** During the COVID-19 era, several restrictions on surgery have been imposed to reduce the infectious risk among patients and staff and further preserve the availability of critical care resources. The aim of the study was to assess their impact on the ophthalmological practice and its medico-legal implications. A retrospective review of electronic medical records of the ophthalmological departments of the University of Cagliari (SGD) and University Magna Græcia of Catanzaro (UMG), from 16 March 2020 to 14 March 2021 (52 weeks), were compared with data from the corresponding period of the previous year. Weekly data on the number and type of diagnoses and procedures performed were collected and analysed in relation to the weekly average of the total number of COVID-19 patients in intensive care units (ICUs) and inpatients in Sardinia and Calabria. Results showed a significant decrease in cataract surgery operations by 47% and 31%, respectively, in the SGD and UMG ($p < 0.05$) during the second semester of the year; this drop occurred at the same time as the increase in COVID-19 patients in ICU and those hospitalised in both regions. Additionally, anterior segment surgery decreased at the UMG by 30% ($p < 0.05$). Vitreoretinal surgery decreased by 27% at the SGD, differently increased amount 31.5% at UMG ($p < 0.05$). The pandemic had a dramatic impact on elective surgery in ophthalmology, quantifying the backlog is the first step in order to understanding the measures to be taken in near future.

**Keywords:** COVID-19; SARS-CoV-2; ophthalmology; lockdown; unlock; second wave; backlog; legal medicine

## 1. Introduction

In December 2019, an infection outbreak of pneumonia of unknown cause attracted global attention to a cluster of patients presenting with pneumonia of unknown origin [1]. Several days later, a new variant of the virus named 'severe acute respiratory syndrome coronavirus 2' (SARS-CoV-2) was identified [2]. In March 2020, it was declared a pandemic, as it had spread to more than one hundred countries [2]. Consequently, the impact of COVID-19 has been impressive and immediate in all the fields of medicine, and all resources were spent on dealing with this pandemic. Given that coronavirus is transmitted by person-to-person contact, via airborne droplets or fomites [3,4], the ophthalmological practice has

been directly and indirectly affected: In fact, in the COVID-19 era, the ophthalmologist has had to cope with the ophthalmological consequences of both SARS-CoV-2 infection and the measures to contain the spread of the pandemic [5–7]. About the first aspect, a recent review by Sen et al. collected numerous studies focused on the manifestations of the infection subdivided and classified for each part of the eye [8]. The authors confirmed that viral conjunctivitis was the most reported ophthalmic manifestation in COVID-19 patients, but other clinical pictures described were also central retinal vein occlusion (CRVO), central retinal artery occlusion (CRAO), paracentral acute middle maculopathy (PAMM), acute macular neuropathy (AMN), vitritis, acute retinal necrosis (ARN), and reactivation of serpiginous choroiditis as isolated reports [8,9].

Concerning the second aspect, measures with the most influence on ophthalmic practice have been the widespread use of face masks and the introduction of smart schooling/working. In June 2020, for the first time, White wrote about the possible detrimental effects of these habits on his blog, coining the acronym 'MADE' (mask-associated dry eye) [10]. A survey administered to 107 healthy students wearing face masks for more than 6 h daily revealed a significant onset of ocular discomfort symptoms that required the use of tear substitutes [11]. Concerning smart schooling, Wang et al. conducted a prospective cross-sectional study using photo-screenings in 123,535 children aged 6 to 13 years from 10 schools in Feicheng, China. The results reported in the age range between 6 and 8 years a substantial myopic shift (approximately −0.3 dioptres [D]) in 2020, compared with the previous 5 years. The authors hypothesised that the refractive status of younger children may be more sensitive to environmental changes than that of older children [12]. To address this issue, a post-pandemic ophthalmological surveillance program should be carried out to detect early myopia shift and possibly treat it, if necessary [13,14].

Since coronavirus disease 2019 (COVID-19) became a pandemic, continuous efforts are being made worldwide to 'flatten the curve' of the contagion. Significant restrictions have been imposed in all countries. On 9 March 2020, the Italian Government imposed unprecedented public health measures to restrict the epidemic by drastically limiting all movement (the 'Stay at Home' Law Decree). Nevertheless, a complete lockdown was started on 22 March 2020 (the 'Closing Italy' Law Decree), and the quarantine progressively came to an end, as of 4 May 2020 [15]. After a summer break, the virus started again to spread, and the second wave of COVID-19 was registered in all European countries in the fall of 2020 [16]. In Italy, the daily incidence of confirmed cases rose slowly from 2 to 3 patients per 100,000 citizens over the month of September and then accelerated rapidly in October, reaching a peak of 58 patients per 100,000 citizens by 13 November [17]. To counter the rapid rise in SARS-CoV-2 infections observed since the end of September, the Italian government progressively increased restrictions again, with the purpose of enhancing physical distancing [18–23]. Between 14 October and 5 November 2020, these interventions were uniformly enacted at the national level. As a result, during the first year of the pandemic, ophthalmologists were required to scale back their routine practice, resulting in a significant reduction in the number of surgical activities.

The aim of this study was to assess the impact of the pandemic containment strategies on surgical activity in two major ophthalmological departments in southern Italy. The burden of COVID-19 in the main surgical categories—namely, anterior segment surgery, cataract surgery, and vitreoretinal surgery—were evaluated and subsequently analysed in relation to the number of new infections, new inpatients, and new intensive care unit (ICU) admissions.

The authors also discussed some of the most important issues from a medico-legal perspective, such as potential sources of professional liability.

## 2. Materials and Methods

### 2.1. Study Design and Settings

This is a multi-centre, observational study performed in two ophthalmological surgery departments in Italy: (1) San Giovanni di Dio (SGD), University of Cagliari, one of the

largest hospitals in Sardinia, an island in southern Italy; (2) Mater Domini, University Magna Græcia of Catanzaro (UMG), the largest tertiary-care ophthalmological departments in Calabria, a region in the southern part of Italy. The surgical activity of both hospitals performed from 16 March 2020 to 14 March 2021, for a total of 52 weeks, was analysed. These data were compared with the analogous data of the previous year, from 18 March 2019 to 15 March 2020 (52 weeks). The study was approved by the local Ethical Committee of both hospitals (Azienda Ospedaliera Università di Cagliari, protocol code PG/2021/19405, date of approval 22/12/2021; University Magna Graecia of Catanzaro, protocol code 322-2021, date of approval 21/10/2021). According to World Health Organisation (WHO) recommendations on the identification and isolation of suspected cases of COVID-19, several containment measures, such as temperature or symptoms check at the hospital entrance and preoperative nasopharyngeal swab, were initiated in both hospitals to prevent access by potentially infected persons during the first year of the pandemic [1]. In addition, all elective procedures and office activities were cancelled in the first quarter of the pandemic and then resumed in a reduced manner. Urgent ophthalmological visits, surgery, and intravitreal injections continued in order to avoid irreversible visual loss.

### 2.2. Data Collection

Data regarding the surgical activity of the ophthalmological department of both hospitals, from 16 March 2020 to 14 March 2021, were collected. This period was referred to as the 'COVID-19 year' and was compared with the analogous data of the previous year, from 18 March 2019 to 15 March 2020, referred to as the 'pre-COVID-19' year. The COVID-19 year was divided into 3 parts according to the national pandemic containment measures: (1) the first period, from 16 March 2020 to 14 June 2020, was the 'lockdown' phase; (2) the second period, from 15 June 2020 to 13 September 2020, was the 'unlock' phase; (3) the third period, from 14 September 2020 to 14 March 2020, was the 'second wave'. The first two phases consisted of 13 weeks each, while the third and final phase consisted of 26 weeks. Similarly, the pre-COVID-19 year were dived into 3 equivalent parts: (1) from 18 March 2019 to 16 June 2019; (2) from 16 June 2019 to 15 September 2019; (3) from 16 September 2019 to 15 March 2020.

Study data were obtained from the database which stores all information obtained from the electronic medical records in both hospitals and segregated into 2 excel sheets using Microsoft Excel 2019 (Microsoft Corporation, Redmond, DC, USA). These data contain patient-specific information about inpatient admissions, diagnosis, and surgical procedures performed. Surgical activities were collected, aggregated on a weekly basis, and divided into 3 categories: (1) anterior segment surgery, (2) vitreoretinal surgery, and (3) cataract surgery. The weekly average of COVID-19 patients in the intensive care unit (ICU) and hospitalised patients in Sardinia and Calabria were noted. Regional data on COVID-19 patients were obtained from the Italian Civil Protection Department and Ministry of Health websites [22,23].

### 2.3. Statistical Analysis

Descriptive statistics were used to elucidate the diagnosis and surgical intervention data using Microsoft Excel®. Continuous variables were tested with the Kolmogorov–Smirnov test for normal distribution. Continuous outcomes were described as means or medians and compared with either a Student's *t*-test or Mann–Whitney test depending on normality. The statistical analysis was performed using SPSS 28.0 for Mac (SPSS Inc., Armonk, NY, USA). Statistical significance was defined as $p < 0.05$.

## 3. Results

### 3.1. Overall Surgery

At the SGD, the total numbers of surgical procedures performed were, respectively, 756 (mean $14.53 \pm 9.88$ procedures per week) during the COVID-19 year, and 1205 (mean $23.17 \pm 12.04$ procedures per week) during the pre-COVID-19 year, with a significant

overall decrease of 37% in the total number of patients during the COVID-19 year ($p < 0.001$) (Figure 1).

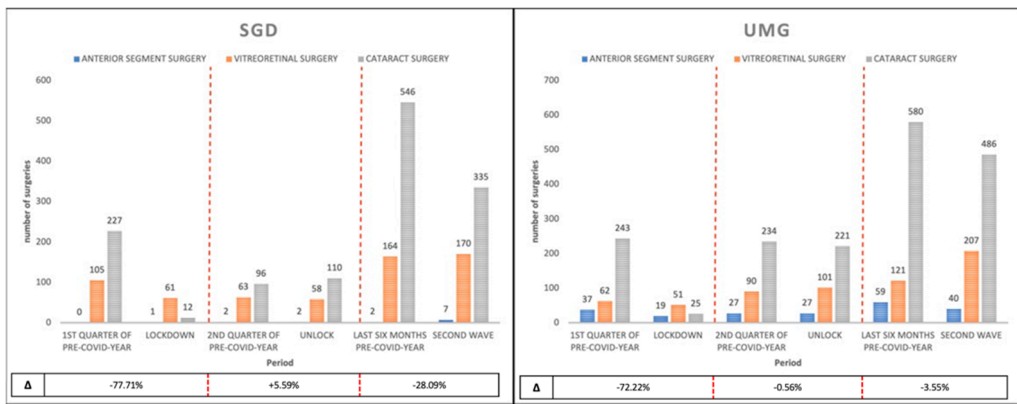

**Figure 1.** The change in the number of surgeries divided into 3 categories—anterior segment surgery (blue), vitreoretinal surgery (orange), and cataract surgery (grey)—performed at San Giovanni di Dio (SGD), University of Cagliari, and Mater Domini, University Magna Græcia of Catanzaro (UMG), in each study period. The box below presents the overall surgery percentage variation during the COVID-19 year.

At the UMG, the total numbers of surgical procedures performed were, respectively, 1177 (mean $22.63 \pm 14.93$ procedures per week) during the COVID-19 year, and 1453 (mean $27.94 \pm 11.30$ procedures per week) during the pre-COVID-19 year, with a significant overall decrease of 19% in the total number of patients during the COVID-19 year ($p < 0.05$) (Figure 1).

The numbers of COVID-19 patients in intensive care units (ICUs) and those hospitalised showed a similar trend in the two regions (Figure 2). Moreover, each peak coincided with the highest reduction of overall surgeries.

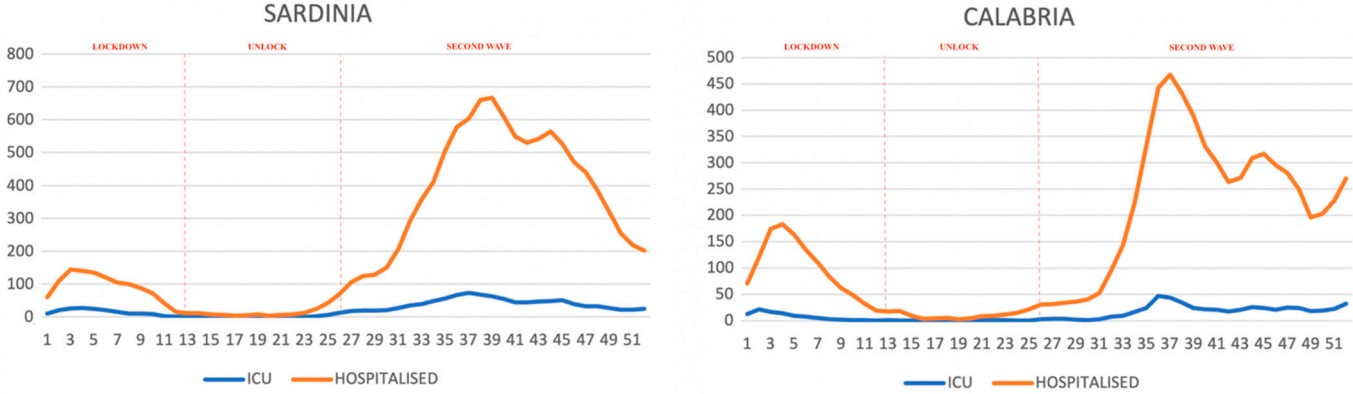

**Figure 2.** The average weekly COVID-19 hospitalisations and intensive care unit (ICU) admissions over time in Sardinia and Calabria during the first year of pandemic (52 weeks). The dotted orange line divided the period studied into three phases: lockdown, unlock, and second wave.

### 3.2. Anterior Segment Surgery

At the SGD, the anterior segment surgeries, performed during the COVID-19 year, included 10 keratoplasties: 1, 2, and 7, respectively, during the lockdown, unlock, and second wave (Table 1). In contrast in the pre-COVID-19 year, four keratoplasties were performed: two in the second quarter and two in the second wave (Table 1). Keratoplasties were performed during the COVID-19 year for the following diagnoses: corneal perforation/rejection (five, 40%), pseudophakic bullous keratopathy (two, 20%), corneal opacity

(two, 20%), and keratoconus (KC) (one, 10%) (Table 2). Differentially, in the pre-COVID-19, three surgeries were performed for corneal perforation/rejection (75%) and one for keratoconus (25%). In both years, penetrating keratoplasty was the only type of corneal transplantation performed (Table 3). Concerning anterior segment surgeries performed at UMG, 86 procedures of corneal transplantation were performed during the COVID-19 year, in contrast to 123 operations during the pre-COVID-19 year. During the COVID-19 year, a significant reduction of 30% was observed ($p < 0.05$). The largest reduction between the 2 years was observed during the lockdown and was 49%, followed by the second wave (32%). Both these trends were statistically significant ($p < 0.05$) (Table 1). In contrast, the same number of surgeries were performed during the unlock phase and the equivalent period of the previous year (n = 27) (Table 1). As shown in Table 2, in the COVID-19 year, the main surgical indications were KC and corneal opacity, respectively, with 32 and 14. A notable decrease was detected for the two main techniques used for lamellar keratoplasty: anterior automated keratoplasty (ALK) and endothelial keratoplasty (EK), respectively, with decreases of 32% and 34%. In comparison, during the COVID-19 year, the PK technique was used more, with an increase of 400%, compared with the pre-COVID-19 year (Table 3).

**Table 1.** Number of anterior segment procedures performed at San Giovanni di Dio (SGD), University of Cagliari, and Mater Domini, University Magna Græcia of Catanzaro (UMG), divided by periods of lockdown, unlock, and second wave, respectively.

| SAN GIOVANNI DI DIO (CAGLIARI) | | | | | | | |
|---|---|---|---|---|---|---|---|
| PRE-COVID-19 YEAR | | | COVID-19 YEAR | | | | |
| Period | N | Median (25–75 Quartiles) | Period | N | Median (25–75 Quartiles) | Δ | *p*-Value |
| First quarter | 0 | 0 [0-0] | Lockdown | 1 | 0 [0-0] | - | 0.25 |
| Second quarter | 2 | 0 [0-0] | Unlock | 2 | 0 [0-0] | 0 | 0.74 |
| Last six months | 2 | 0 [0-0] | Second Wave | 7 | 0 [0-1] | +71.42% | 0.79 |
| Total | 4 | | Total | 10 | | +60% | 0.23 |
| MATER DOMINI (CATANZARO) | | | | | | | |
| PRE-COVID-19 YEAR | | | COVID-19 YEAR | | | | |
| Period | N | Weekly Mean ± SD | Period | N | Weekly Mean ± SD | Δ | *p*-Value |
| First quarter | 37 | 2.85 ± 0.99 | Lockdown | 19 | 1.46 ± 1.76 | −48.64% | *0.02* |
| Second quarter | 27 | 2.08 ± 1.55 | Unlock | 27 | 2.08 ± 1.93 | 0% | 1 |
| Last six months | 59 | 2.27 ± 1.28 | Second Wave | 40 | 1.54 ± 1.33 | −32.20% | *0.04* |
| Total | 123 | | Total | 86 | | −30.08% | *0.02* |

N: number of procedures; Mann–Whitney U test. Data are given in median (quartile 25–75); statistically significant values are in italics.

**Table 2.** Comparison of different diagnoses of surgically treated anterior segment diseases at San Giovanni di Dio (SGD), University of Cagliari, and Mater Domini, University Magna Græcia of Catanzaro (UMG).

|  | SAN GIOVANNI DI DIO (CAGLIARI) | | | MATER DOMINI (CATANZARO) | | |
|---|---|---|---|---|---|---|
| Diagnosis | PRE-COVID-19 | COVID-19 | Δ | PRE-COVID-19 | COVID-19 | Δ |
| KERATOCONUS | 1 | 1 | 0% | 44 | 32 | −27% |
| CORNEAL OPACITY | - | 2 | 200% | 20 | 14 | −30% |
| FUCHS' ENDOTHELIAL DISTROPHY | - | - | - | 6 | 8 | 33% |
| PSEUDOPHAKIC BULLOUS KERATOPATHY | - | 2 | 200% | 24 | 10 | −58% |
| ECTASIA IN CORNEAL GRAFT | - | - | - | 9 | 8 | −11% |
| CORNEAL GRAFT REJEC-TION/PERFORATION | 3 | 5 | 40% | 20 | 14 | −67% |

**Table 3.** Comparison of different types of anterior segment surgical procedures at San Giovanni di Dio (SGD), University of Cagliari, and Mater Domini, University Magna Græcia of Catanzaro (UMG).

|  | SAN GIOVANNI DI DIO (CAGLIARI) | | | MATER DOMINI (CATANZARO) | | |
|---|---|---|---|---|---|---|
| Surgical Procedure | PRE-COVID-19 | COVID-19 | Δ | PRE-COVID-19 | COVID-19 | Δ |
| PK | 4 | 10 | 60% | 1 | 5 | 400% |
| ALK | - | - | - | 57 | 39 | −32% |
| EK | - | - | - | 38 | 25 | −34% |
| MUSHROOM KERATOPLASTY | - | - | - | 16 | 6 | −63% |
| GRAFT REVISION | - | - | - | 11 | 11 | 0% |

### 3.3. Vitreoretinal Surgery

Overall, 289 surgical procedures for diseases affecting the posterior segment of the eye were performed during the COVID-19 year at the SGD. These were distributed as follows: 61 during the lockdown (weekly mean $4.69 \pm 2.81$), 58 during the unlock phase (weekly mean $4.46 \pm 3.15$), and 170 during the second wave (weekly mean $6.53 \pm 3.38$). During the pre-COVID-19 year, a total of 332 surgeries were carried out: in the first quarter, 105 procedures were performed (weekly mean $8.07 \pm 3.27$), whereas in the second quarter, 63 operations (weekly mean $4.84 \pm 3.95$) and in the second wave, 164 operations were performed (weekly mean $6.30 \pm 2.82$) (Table 4). Therefore, a reduction in surgical activity was registered during the lockdown and unlock, compared with the same periods of the pre-COVID-19 year, by 41.9% ($p < 0.01$) and 7.94%, respectively, whereas an increase of 3.66% in operations was found during the second wave, compared with the previous year. Table 5 represents the cause-specific distribution and the type of surgery performed, respectively, during COVID-19 and the pre-COVID-19 years.

**Table 4.** Number of vitreoretinal surgical procedures performed at San Giovanni di Dio (SGD), University of Cagliari, and Mater Domini, University Magna Græcia of Catanzaro (UMG), divided by periods of lockdown, unlock, and second wave, respectively.

| SAN GIOVANNI DI DIO (CAGLIARI) | | | | | | | |
|---|---|---|---|---|---|---|---|
| PRE-COVID-19 YEAR | | | COVID-19 YEAR | | | | |
| Period | N | Weekly Mean ± SD | Period | N | Weekly Mean ± SD | Δ | *p*-Value |
| First quarter | 105 | 8.07 ± 3.27 | Lockdown | 61 | 4.69 ± 2.81 | *−41.9%* | *0.009* |
| Second quarter | 63 | 4.84 ± 3.95 | Unlock | 58 | 4.46 ± 3.15 | −7.94% | 0.78 |
| Last six months | 164 | 6.30 ± 2.82 | Second Wave | 170 | 6.53 ± 3.38 | +3.66% | 0.79 |
| Total | 332 | | Total | 289 | | −27.26% | 0.2 |
| MATER DOMINI (CATANZARO) | | | | | | | |
| PRE-COVID-19 YEAR | | | COVID-19 YEAR | | | | |
| Period | N | Weekly Mean ± SD | Period | N | Weekly Mean ± SD | Δ | *p*-Value |
| First quarter | 62 | 4.77 ± 2.65 | Lockdown | 51 | 3.92 ± 2.84 | −17.74% | 0.44 |
| Second quarter | 90 | 6.92 ± 4.05 | Unlock | 101 | 7.76 ± 3.91 | +12.22% | 0.59 |
| Last six months | 123 | 4.65 ± 3.00 | Second Wave | 207 | 7.96 ± 5.28 | *+71.07%* | *0.007* |
| Total | 273 | | Total | 359 | | *+31.50%* | *0.04* |

N: number of procedures; statistically significant values in italics.

**Table 5.** Comparison of different diagnoses of surgically treated posterior segment diseases at San Giovanni di Dio (SGD), University of Cagliari, and Mater Domini, University Magna Græcia of Catanzaro (UMG).

| Diagnosis | SAN GIOVANNI DI DIO (CAGLIARI) | | | MATER DOMINI (CATANZARO) | | |
|---|---|---|---|---|---|---|
| | PRE-COVID-19 | COVID-19 | Δ | PRE-COVID-19 | COVID-19 | Δ |
| RETINAL DETACHMENT | 123 | 112 | −9% | 91 | 148 | 63% |
| VITREOUS HEMORRHAGE | 20 | 30 | 50% | 36 | 49 | 36% |
| EPIRETINAL MEMBRANE | 59 | 34 | −42% | 40 | 45 | 13% |
| SILICONE OIL IN VITREOUS CAVITY | 13 | 21 | 62% | 33 | 39 | 18% |
| RECURRENT RETINAL DETACHMENT | 45 | 35 | −22% | 20 | 26 | 30% |
| MACULAR HOLE | 44 | 24 | −45% | 12 | 20 | 67% |
| LENS FRAGMENTS IN VITREOUS CAVITY | 5 | 9 | 80% | 18 | 13 | −28% |
| IOL DISLOCATED IN VITREOUS CAVITY | 8 | 10 | 25% | 18 | 12 | −33% |
| DIABETIC RETINOPATHY | 10 | 11 | 10% | 3 | 5 | 67% |
| SYNCHYSIS SCINTILLANS | 0 | 1 | - | 0 | 1 | - |
| IOL OPACIFICATION | - | - | - | 1 | 1 | 0% |
| VITRITIS | - | - | - | 3 | 0 | −100% |
| ENDOPHTHALMITIS | 1 | 0 | - | - | - | - |
| OCULAR TRAUMA | 4 | 112 | −50% | - | - | - |

The activity of the UMG vitreoretinal surgical unit significantly increased by 31% during the pandemic ($p < 0.05$). In particular, 359 surgical interventions for posterior segment diseases were performed during the COVID-19 year, and they were distributed as follows: 51 during the lockdown phase (weekly mean $3.92 \pm 2.84$), 101 during the unlock (weekly mean $7.76 \pm 3.91$), and 207 during the second wave (weekly mean $7.96 \pm 5.28$). Conversely, during the pre-COVID-19 year, a total of 273 surgeries were carried out: 62 procedures in the first quarter (weekly mean $4.77 \pm 2.65$), 90 operations in the second quarter (weekly mean $6.92 \pm 4.05$), and 123 during the second wave (weekly mean $4.65 \pm 3.00$). The highest number of interventions were performed during the second wave, corresponding to 71% of the total ($p < 0.05$). Table 5 represents the cause-specific distribution and the type of surgery performed, respectively, during COVID-19 and the pre-COVID-19 years. Tables 5 and 6 represent the cause-specific distribution and the type of surgery performed, respectively, during COVID-19 and the pre-COVID-19 years.

**Table 6.** Comparison of different types of vitreoretinal surgical procedures at San Giovanni di Dio (SGD), University of Cagliari, and Mater Domini, University Magna Græcia of Catanzaro (UMG).

| Surgical Procedure | SAN GIOVANNI DI DIO (CAGLIARI) | | | MATER DOMINI (CATANZARO) | | |
|---|---|---|---|---|---|---|
| | PRE-COVID-19 | COVID-19 | Δ | PRE-COVID-19 | COVID-19 | Δ |
| PPV | 308 | 264 | −14% | 214 | 307 | 43% |
| PPV + SCLERAL FIXATED IOL | 14 | 17 | 21% | 30 | 23 | −23% |
| SCLERAL BUCKLING | 10 | 8 | −20% | 29 | 29 | 0% |

### 3.4. Cataract Surgery

The number of cataract surgeries decreased by 47% and 31%, respectively, in the SGD and UMG (Table 7). These surgeries, if not urgent, were postponed in the first 13 weeks of the pandemic, and as a result, a statistically significant decrease was observed (SGD by 95% and UMG by 89%, $p < 0.001$). During the unlock phase, efforts were made in both hospitals to recover backlogs: at the SGD, the activity was increased by 14%, compared with the second quarter of the pre-COVID-19 year ($p < 0.001$), whereas at the UMG, the activity was still slightly inferior, compared with the equivalent period of the previous year (5%) (Table 7).

**Table 7.** Number of cataract surgical procedures performed at San Giovanni di Dio (SGD), University of Cagliari, and Mater Domini, University Magna Græcia of Catanzaro (UMG), divided by periods of lockdown, unlock, and second wave, respectively.

| SAN GIOVANNI DI DIO (CAGLIARI) | | | | | | | |
|---|---|---|---|---|---|---|---|
| PRE-COVID-19 YEAR | | | COVID-19 YEAR | | | | |
| Period | N | Weekly Mean ± SD | Period | N | Weekly Mean ± SD | Δ | *p*-Value |
| First quarter | 227 | $17.46 \pm 8.77$ | Lockdown | 12 | $0.92 \pm 1.11$ | −94.71% | *<0.001* |
| Second quarter | 96 | $7.30 \pm 6.27$ | Unlock | 110 | $8.46 \pm 6.13$ | +14.58% | *<0.001* |
| Last six months | 546 | $21 \pm 9.96$ | Second Wave | 335 | $12.88 \pm 8.29$ | −38.64% | 0.66 |
| Total | 869 | | Total | 457 | | −47.41% | *0.002* |
| MATER DOMINI (CATANZARO) | | | | | | | |
| PRE-COVID-19 YEAR | | | COVID-19 YEAR | | | | |
| Period | N | Weekly Mean ± SD | Period | N | Weekly Mean ± SD | Δ | *p*-Value |
| First quarter ¥ | 243 | $18.69 \pm 6.86$<br>19 (16–23) | Lockdown ¥ | 25 | $1.92 \pm 4.03$<br>1 (0–2) | −89.71% | *<0.001* |
| Second quarter | 234 | $18 \pm 10.71$ | Unlock | 221 | $17 \pm 11.87$ | −5.56% | 0.82 |
| Last six months | 580 | $22.31 \pm 9.12$ | Second Wave | 486 | $18.69 \pm 12.95$ | −16.21% | 0.24 |
| Total | 1057 | | Total | 732 | | −30.75% | *0.07* |

N: number of procedures; statistically significant values in italics, ¥ Mann–Whitney U test. Data are given in median (quartile 25–75).

## 4. Discussion and Medico-Legal Implications

The SARS-CoV-2 pandemic has imposed extraordinary efforts and restriction measures to flatten the virus spread curve. Since the human transmission of the COVID-19 occurs primarily through droplets, contacts, and fomites, social distancing measures were imposed in all countries [24]. In March 2020, in view of a health emergency, the Ministry of Health and the National Government recommended the public use the hospitals only in case of an absolute emergency [23]. Consequently, the Medical Office of several hospitals required the minimisation of elective surgical activity and deployed forces to cope with the exponential increase in infected and newly hospitalised patients. Both Sardinia and Calabria have adopted these measures [25,26]. In this context, since the second week of March 2020, outpatient activities and non-urgent admissions at San Giovanni di Dio (University of Cagliari, Sardinia) and Mater Domini (University Magna Græcia of Catanzaro, Calabria) have been suspended [27–30]. These measures remained in use until mid-June 2020. On 11 June 2020, after a drastic reduction in the number of infected people, the Italian First Minister signed the document governing the reopening of most activities, thus beginning the third phase of the emergency [31]. Concomitantly, elective surgical and outpatient activities at Sardinia and Calabria Hospitals have been restarted [32]. This phase remained in place until mid-September. The second wave of the pandemic coincided with the reopening of schools with new and significant outbreaks in Italy's largest cities [33]. New restrictions were imposed due to the resulting increase in infections, but this time, no longer nationally but regionally, based on the increase in infections and reduction in ICU beds. As a result, three phases were recognised in the first year of the COVID-19 pandemic. The first, from mid-March to mid-June, coincides with the National Lockdown; the second phase, the unlock phase, runs from the second half of June until the first half of September; lastly, the third and final phase known was the second wave until March 2021. We presently discussed the impact of the first year of the national and regional measures of virus spread containment in ophthalmology units of two of the major hospitals in southern Italy hospitals: San Giovanni di Dio in Sardinia and Mater Domini in Calabria. The main types of surgical procedures performed in the two hospitals were corneal surgery, vitreoretinal surgery, and cataract surgery. To the best of our knowledge, the present study is the first of its kind depicting the results of one year of measures of national containment on the ophthalmological surgical activity.

During the lockdown, in both hospitals, a great reduction in surgical activity was observed, compared with the surgical activity during the equivalent period of the previous year. A reduction of 30% in all surgical services was observed in both ophthalmological units (Figure 1). With a closer examination regarding the subtype of surgery, it is evident that cataract surgery was more affected, with a drop of around 90% in both hospitals (Table 7). A similar trend has been described by Das et al. [34] in a cross-sectional, observational study that included patients who underwent cataract surgery, presenting between 23 March 2019 and 31 March 2021 to 4 tertiary and 20 secondary centres of a multi-tier ophthalmology network located in India. The authors found a drop of 92.5% in the number of cataract surgeries in the lockdown phase. This result was explained by both the containment measures imposed by the Indian government, as well as by the guidelines released by the All-India Ophthalmological Society (AIOS) [35]. Our findings suggest that the number of cataract surgical procedures recovered in the unlock phase, reaching its peak between July and August (Figure 3E,F). The surgical activity, compared with that during the equivalent previous year, increased by 14.58% at the SGD, whereas at the UMG, the activity was slightly lower (−5.56%) (Table 7). During the second wave, the number of cataracts fell again by about 38% and 30% in the SGD and UMG, respectively (Table 7). These fluctuating trends do not agree with the findings described by Das et al. [34]. In this study, the authors found that the number of cataract surgeries recovered from May 2020, increasing steadily to exceed the monthly average by August 2020. In our case, this trend was a consequence of the enormous increase in COVID-19 patients and the progressive reduction in ICU beds. Indeed, if during the lockdown, southern Italy was less affected

by the pandemic, this scenario was not repeated in the second wave. It could be assumed that this situation led to panic among elderly patients. As reported in several articles, the delay or postponement of cataract surgery has negative impacts on the elderly health (e.g., fractures due to falls, difficulty in driving), thus predisposing them to traffic accidents, depression, and dementia [35–37].

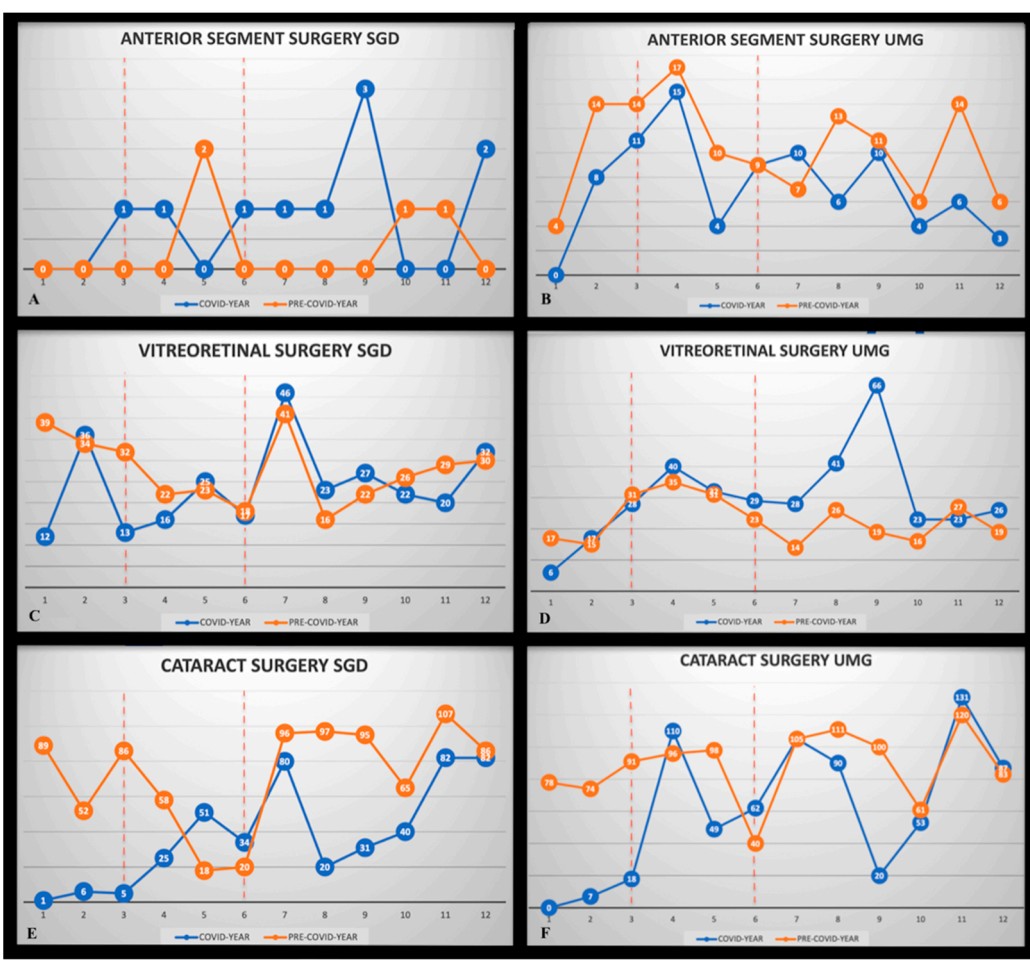

**Figure 3.** The monthly number of surgical procedures performed during the COVID-19-year (blue) compared with pre-COVID-19 year (orange) for the different categories: (**A**) corneal surgery performed at San Giovanni di Dio (SGD); (**B**) corneal surgery performed at University Magna Græcia of Catanzaro (UMG); (**C**) vitreoretinal surgery performed at SGD; (**D**) vitreoretinal surgery performed at (UMG); (**E**) cataract surgery performed at SGD; (**F**) cataract surgery performed at UMG. The dotted orange line in each figure divided the period studied into three phases: lockdown, unlock, and second wave, from left to right, respectively.

Concerning corneal surgeries, similar results were observed. Corneal blindness is the third leading cause of blindness in the world [38], and corneal transplantation remains the only technique able to restore tissue transparency [39]. However, apart from a small percentage of cases (i.e., corneal perforation or infection), corneal transplants are, in general, elective surgeries [40]. It is important to note that, as with most solid organ transplants, the risk of transmitting infectious diseases in the recipient or those handling donor tissue must always be taken into account [41,42]. As a result of these considerations, the corneal surgical activity decreased due to the reorganisation of the national health service and initial uncertainties related to tissue procurement and storage [43,44]. As reported in a recent Italian overview that analysed data from the Italian Society of Eye Banks (SIBO) during the first quarter of the pandemic, a significant reduction of around 60% was observed [40]. At the UMG, similarly, a significant dramatic fall in total corneal surgeries during the

lockdown period (49%, *p* < 0.05), followed by a return to normal in the unlock phase, was found (Table 1). Unfortunately, a second decrease of about 32% was observed in the second wave (*p* < 0.05) (Figure 3B). Finally, analysing the types of corneal transplants performed, our results confirmed the trend reported by Aiello et al. [40], with a reduction of around 30–40%, for both anterior and endothelial keratoplasty; however, in contrast to their results, we observed an increase in PK. Different results were collected at the SGD, but this is due to the fact that, in both years, only emergency surgery was performed, as confirmed by the diseases treated (Table 2).

Differentially from the surgical services described above, vitreoretinal surgical service was also affected by the measure for the pandemic containment but, in our case, in the opposite direction. There is poor information regarding the epidemiology of vitreoretinal emergencies during the lockdown [45], but in accordance with the general trend, a downsizing of vitreoretinal procedures was also reported. For example, the frequency of urgent or emergent vitreoretinal procedures substantially decreased in the US during the lockdown [46] and in India [47]. A similar scenario was observed at the SGD and UMG during the lockdown, but the trend reversed over the remaining 9 months (Figure 3C,D). The principal reason is that even in the worst months during the pandemic, the emergency eye-care centre of both clinics did not interrupt its service. As a result, patients affected by a retinal detachment, diabetic retinopathy, lens fragment in the vitreous chamber, or other emergencies were promptly referred for surgery (Table 5). Furthermore, the increase in the number of vitreoretinal surgeries during the second wave, more evident in Calabria, could be the consequence of the shut-down of local smaller hospitals and private clinics. In order to provide and preserve the continuity of care, various measures were adopted in accordance with the literature—first, the creation of the deferrable/non-deferrable patient list; second, a combined surgery including phacoemulsification; and finally, the choice of using endotamponade to reduce the follow-up visits and the risk of recurrence [32,48].

The results obtained showed a fluctuating trend regarding the number of patients treated during the period examined. With regard to the emergencies for which treatment could not be postponed, the numbers did not change with respect to the pre-pandemic period; however, the same cannot be affirmed with regard to elective surgery, with a clinical impact still not completely clear. Some authors have actually indicated that one of the 'collateral damages' of the pandemic is precisely the impact it has had, directly and indirectly, on the management of other morbid conditions [49,50]. Healthcare facilities have had to balance the need to ensure all care demands, on the one hand, and the need to minimise the risk of infection for patients and caregivers, on the other. Clearly, some ethical and medico-legal implications might arise from such issues [51]. In detail, there are two aspects that could represent a source of professional liability: On the one side, the postponement of therapeutic procedures could lead to an aggravation of the same pathology, with potential damage to the patient, and, on the other side, the risk that a patient could contract the virus during hospitalisation, which could be a cause of liability for nosocomial infection, especially in case of non-compliance with protocols and guidelines. In fact, although judicial data on med mal cases in the pandemic context are not yet available, a drastic increase in medical malpractice claims has been reported against individual professionals or, more generally, against healthcare facilities, in relation, for example, to organisational and inadequate medical assistance in non-COVID emergency diseases and the drastic increase in deaths in nursing homes for the elderly [52]. These events could also involve the ocular surgeon, also considering that, according to a recently published study referring to the pre-pandemic period, ophthalmology is, in Italy, the fourth most involved specialist branch in civil liability judgments, with a ratio of 50% between no. of cases and no. of convictions and an average paid compensation of approximately EUR 42.266,00 [53]. With reference to the more strictly regulatory aspects, Law no. 76 of 28.05.2021, containing urgent measures for the containment of the SARS-CoV-2 epidemic with regard to vaccinations against SARS-CoV-2, introduced the so-called 'criminal shield' in the Italian legal system for the crimes of manslaughter (Article 589 of the Criminal Code)

and culpable personal injury (Article 590 of the Criminal Code) that have found their cause in the epidemiological emergency context. This new legislation also introduced the criteria on which the judge will be called upon to assess the existence of gross negligence, the only hypothesis from which the liability referred in the aforementioned provisions of the Criminal Code may be derived. In this sense, among the conditions to be taken into account are the limited scientific knowledge at the time of the event on SARS-CoV-2 pathologies and appropriate therapies, the scarcity of human and material resources actually available in relation to the number of cases to be treated, and the lesser degree of experience and technical knowledge possessed by non-specialised personnel to deal with the emergency. Therefore, if it is possible to say that the legislator has increased the level of protection for health professionals in the criminal field, the situation in the civil sector seems far from simple. To date, in fact, no such provision has been made for protection even in the civil law context of compensation for damage, where there are substantially different criteria from those of the criminal sphere; for example, a different concept of proof of the causal link governs these different areas of law. It is, therefore, evident that even in a pandemic context such as the one under discussion, in the absence of further regulatory changes, the ordinary rules of civil liability continue to apply. In particular, the distinction between the liability of the healthcare facility and the liability of the healthcare professional takes on partially different characteristics in Italy. This distinction places the healthcare facility on a different footing from the individual healthcare professional, representing a kind of guarantee figure with respect to the protection of citizens' health, especially in a pandemic context.

Although the pandemic has had a significant impact on surgery rates and outpatient services worldwide, it had also some positive implications. Firstly, the governments realised the importance of an efficient public healthcare system. Secondly, more importance is given to the role of telemedicine which minimises the need for physical interaction and allows territorial decentralisation of specialised healthcare services [54,55]. We believe that teleconsultation services could be part of regular care in the near future to minimise hospital visits, quantify the burden following a pandemic, and finally discern which cases will be the most urgent from the less urgent ones.

## 5. Conclusions

In conclusion, we presented herein our experience on the impact of the COVID-19 pandemic on the surgical services of two of the largest ophthalmological units in southern Italy. In the first year of the COVID-19 pandemic, a significant decrease in the surgical volume was registered during the lockdown period, which gradually decreased during the following 9 months, compared with the pre-COVID-19 year. Quantifying the backlog caused by COVID-19 is the first step in understanding the direction of health measures to be taken in the coming years. However, the impact of the second year of the COVID-19 pandemic still remains to be evaluated.

**Author Contributions:** Conceptualisation, G.D. and D.M.; methodology, G.D. and D.M.; formal analysis, G.D. and F.T.; investigation, G.D., D.M., A.V., F.G., O.S., F.T., M.B., A.C. and L.L.; data curation, G.D., F.T., A.M. and M.B.; writing—original draft preparation, G.D. and D.M.; writing—review and editing, G.G., E.P. and V.S.; supervision, G.G. All authors have read and agreed to the published version of the manuscript.

**Funding:** This research received no external funding.

**Institutional Review Board Statement:** The study was approved by the Ethics Committee of Azienda Ospedaliera Università di Cagliari (protocol code PG/2021/19405, date of approval 22/12/2021) and University Magna Graecia of Catanzaro (protocol code 322-2021, date of approval 21/10/2021).

**Informed Consent Statement:** Not applicable.

**Data Availability Statement:** Not applicable.

**Conflicts of Interest:** The authors declare no conflict of interest.

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
