# Peer review of "Effect of the First Year of COVID-19 Pandemic on Ophthalmological Practice: A Multi-Centre Italian Study with a Focus on Medico-Legal Aspects"

_applsci, doi:10.3390/app12094453_

Round 1
Reviewer 1 Report
Dear authors, the manuscript is interesting however there are several issues with statistics/methods. Please find below some indications:
- In the text there are discrepancies in the “Covid-19” terms, please use always the same term (COVID-19 in capital letters)
- Study design and methods: I suggest removing "retrospective" although the data collection is retrospective
- Please consider that Declaration of Helsinki refers to medical research involving human subjects
- Based on background and study aim, it is not clear why the authors describe the following procedure: “Patients requiring visits were screened in checkpoints placed at each hospital entrance for the evaluation of body temperature, respiratory symptoms, and history of suspected contacts. With a tempera-ture above 37.5°C, respiratory symptoms, or positive anamnesis for contact with infected or suspected individuals, hospital access was denied up the performance of a nasopha-ryngeal swab. Patients requiring department admission or surgery were screened for se-vere acute respiratory coronavirus 2 (SARS-CoV- 2) with a nasopharyngeal swab for re-verse transcription polymerase chain reaction test. All persons were invited to wear a new surgical mask. All elective procedures and office activities were cancelled in the first quar-ter of the pandemic and then resumed in a reduced manner. Urgent ophthalmological visits, surgery and intravitreal injections continued in order to avoid irreversible visual loss.”
- 3 statistical Analysis: in descriptive statistics are not described techniques used to summarize qualitative data. Authors tested normality of data and used Mann-Whitney test, is it correct that there are only means and standard deviations? The “statistical significance” information is present twice (remove the one in the sixth row). About linear regression it is not adequate to study the following outcome “ the number of … procedures”.
- Results
- figure 1 consider to add measure unite.
- Table 1 shows in the title “average number” but in table refers to Cagliari there are not data (what is N? mean and sd? ).
- Table 8: why authors showed R2 for linear regression analyses? There are not information about coefficients? From this table is not possible to evaluate relationship about outcome and covariates. Outcome variables is not clear (measure unit).
Author Response
Dear authors, the manuscript is interesting however there are several issues with statistics/methods. Please find below some indications:
Point 1: In the text there are discrepancies in the “Covid-19” terms, please use always the same term (COVID-19 in capital letters)
Response 1: Thank you for your comment. Correction has been made in the revised manuscript.
Point 2: Study design and methods: I suggest removing "retrospective" although the data collection is retrospective
Response 2: Correction has been made in the revised manuscript.
Point 3: Please consider that Declaration of Helsinki refers to medical research involving human subjects
Response 3: Thank you for your comment. Correction has been made in the revised manuscript.
Point 4: Based on background and study aim, it is not clear why the authors describe the following procedure: “Patients requiring visits were screened in checkpoints placed at each hospital entrance for the evaluation of body temperature, respiratory symptoms, and history of suspected contacts. With a temperature above 37.5°C, respiratory symptoms, or positive anamnesis for contact with infected or suspected individuals, hospital access was denied up the performance of a nasopharyngeal swab. Patients requiring department admission or surgery were screened for severe acute respiratory coronavirus 2 (SARS-CoV- 2) with a nasopharyngeal swab for re-verse transcription polymerase chain reaction test. All persons were invited to wear a new surgical mask. All elective procedures and office activities were cancelled in the first quarter of the pandemic and then resumed in a reduced manner. Urgent ophthalmological visits, surgery and intravitreal injections continued in order to avoid irreversible visual loss.”
Response 4: Thank you for your comment. We thought that the description of restriction measures imposed by our hospital could be useful for the readers to understand how was the hospital access during the lockdown period. Following your suggestion, we removed the section about all the measures adopted by each hospital and we just maintained the sentence “In addition, all elective procedures and office activities were cancelled in the first quarter of the pandemic and then resumed in a reduced manner. Urgent ophthalmological visits, surgery and intravitreal injections continued in order to avoid irreversible visual loss.”
Point 5: 3 statistical Analysis: in descriptive statistics are not described techniques used to summarize qualitative data. Authors tested normality of data and used Mann-Whitney test, is it correct that there are only means and standard deviations? The “statistical significance” information is present twice (remove the one in the sixth row). About linear regression it is not adequate to study the following outcome “ the number of … procedures”.
Response 5: Thank you for pointing it out. Continuous outcomes were described as means or medians and compared with either a 2-sided Student t test or Mann-Whitney test depending on normality. Correction has been made in the revised manuscript. (Line 185-187). The median and the quartile (25, 75) were added to the table and the data examined with Mann-Whitney were highlighted with the symbol ¥ as shown in the description of the tables. We better explained in the statistical analysis section how we analyzed data with simple linear regression model.
Point 6: Results
figure 1 consider to add measure unite.
Response 6: Thank you for your comment. Correction has been made in the revised manuscript.
Point 7: Table 1 shows in the title “average number” but in table refers to Cagliari there are not data (what is N? mean and sd?).
Response 7: Thank you for pointing this out. Following your suggestion, we added 2 columns for each examined period in Table 1 with Median and quartile (25, 75).
Point 8: Table 8: why authors showed R2 for linear regression analyses? There are not information about coefficients? From this table is not possible to evaluate relationship about outcome and covariates. Outcome variables is not clear (measure unit).
Response 8: We appreciate your comment. We modified the Table 8, adding the coefficient R. The measure unit was explained in the table description.
Reviewer 2 Report
A very interesting manuscript linking COVID-19 science, medical practice and law.
Some general and specific comments:
- It is more than two years since the start of this evolving pandemic. COVID-19 is part of our daily life - could consider abbreviating the introductory paragraph in Introduction or merging with the second paragraph, to give a proper focus on Ophthmology practice during COVID-19.
- The paper presented many extensive tables to illustrate the change in procedure numbers. The data could be more effectively presented graphically (similar to Figure 3). I would also suggest pre-COVID-19 data to be placed on the left column and COVID-19 period data be placed on the right column to respect the usual way of charting things by temporal sequence.
- The diagnoses / procedures in Table 2 and Table 3 have many "sparse categories" with no data for Cagliari. Proper grouping of diagnoses / procedures would allow a more meaningful comparison (not an infinite % increase one hospital and a decrease in another).
- Could instead consider a line graph for Figure 1.
I went into the article looking for an in-depth discussion on the medico-legal issues pertaining to Ophthalmology practice in Italy during the pandemic. I would have expected statistics on complaints and example civil lawsuits against the hospitals / Regional Health Authority / doctors arising from delays and hospital-acquired infections. On medico-legal aspect there is also staff and HR issues that must be addressed.
This is an overall descriptive study with useful information about Ophthalmology practice in Italy during this difficult period. The authors could consider more extended and structured discussion on the legal issues, and appropriately shortening the somewhat lengthy descriptive paragraphs in 3. Medico-legal observations should be added to 3 to keep the final part manageable in length.
The figures need to be upgraded to hi-res versions. Please do not just copy and paste presentation slides as the slide effects and shading could make viewing difficult, and the text in the figures would appear blurred.
Author Response
Point 1: could consider abbreviating the introductory paragraph in Introduction or merging with the second paragraph, to give a proper focus on Ophthalmology practice during COVID-19.
Response 1: We appreciate your comment. Following your suggestion, we have reduced the introductory paragraph and merged the first two paragraphs.
Point 2: The paper presented many extensive tables to illustrate the change in procedure numbers. The data could be more effectively presented graphically (similar to Figure 3). I would also suggest pre-COVID-19 data to be placed on the left column and COVID-19 period data be placed on the right column to respect the usual way of charting things by temporal sequence.
Response 2: Thank you for your comment, we have assumed that the graphical presentations in Figure 3 are comprehensive to show the change in the number of procedures. The data in the other tables, on the other hand, are useful for showing the significantly statistical reduction in surgical procedures. We followed your suggestion and reversed the COVID-19 and pre-COVID-19 data in all tables.
Point 3: The diagnoses / procedures in Table 2 and Table 3 have many "sparse categories" with no data for Cagliari. Proper grouping of diagnoses / procedures would allow a more meaningful comparison (not an infinite % increase one hospital and a decrease in another).
Response 3: Thank you for pointing this out. We have reduced the "sparse categories" to give a better understanding of the data
Point 4: Could instead consider a line graph for Figure 1.
Response 4: Thank you for your consideration. We preferred to use the bar graph because in this way we supposed the reduction is clearer to readers. We reversed COVID-19 and pre-COVID-19 data to respect the way of chart by temporal sequence.
Point 5: I went into the article looking for an in-depth discussion on the medico-legal issues pertaining to Ophthalmology practice in Italy during the pandemic. I would have expected statistics on complaints and example civil lawsuits against the hospitals / Regional Health Authority / doctors arising from delays and hospital-acquired infections. On medico-legal aspect there is also staff and HR issues that must be addressed. This is an overall descriptive study with useful information about Ophthalmology practice in Italy during this difficult period. The authors could consider more extended and structured discussion on the legal issues, and appropriately shortening the somewhat lengthy descriptive paragraphs in 3. Medico-legal observations should be added to 3 to keep the final part manageable in length.
Response 5: Thank you for your consideration, we have shortened the descriptive paragraph in 3 (particularly 3.2 and 3.3). For what concerns the data relating to civil lawsuits introduced and deriving from covid-related malpractice, published reference cases are not yet available, since it must be considered that, in Italy, the statute of limitations is 10 years if the healthcare facility is sued and 5 years if the doctor is prosecuted; moreover, even with regard to any lawsuits that have already been introduced, there are latency periods that must be considered between the introduction of the lawsuit and the outcome of the judgement. Therefore, it is likely that the effects at judicial level will be appreciable only in the near future.
Point 6: The figures need to be upgraded to hi-res versions. Please do not just copy and paste presentation slides as the slide effects and shading could make viewing difficult, and the text in the figures would appear blurred.
Response 6: We improved the quality of images.
Reviewer 3 Report
COVID-19 has impacted different aspects of human health. Due to the pandemic control measures adopted around the world, including restricting access to elective medical surgeries, there was a huge impact on people's health. In this work, the authors evaluated the impact of pandemic containment strategies on surgical activity in two large ophthalmic departments in southern Italy. The authors conducted a multicenter retrospective observational study comprising data from March 16, 2020, to March 14, 2021, the first year of the pandemics. The data were compared with the same period of the previous year.
The authors compiled several data and analyzed them considering three different moments, named “lockdown”, “unlock” and “second wave”. A series of statistical analyzes were performed to allow a proper interpretation of the data. The authors show evidence of the impact of COVID-19 in decreasing surgical volume compared to the pre-pandemic period, and discuss how this may impact human health and what the medical community should consider in the event of a future health crisis.
The study was well designed and the results are solid with good discussion. However, this manuscript is too specific to be published in a journal like Applied Science and I believe it would fit better in a journal focused on ophthalmology.
That said, I suggest the authors improve the quality of figure 3. The resolution is very low, which makes it difficult to read the numbers in the graphs. Also, on page 12 the authors mention table 10, but it is not shown. I believe they refer to table 8. If so, this should be corrected. Finally, there are some typos throughout the text that should be corrected. A careful check will resolve the issue.
Author Response
Point 1: COVID-19 has impacted different aspects of human health. Due to the pandemic control measures adopted around the world, including restricting access to elective medical surgeries, there was a huge impact on people's health. In this work, the authors evaluated the impact of pandemic containment strategies on surgical activity in two large ophthalmic departments in southern Italy. The authors conducted a multicenter retrospective observational study comprising data from March 16, 2020, to March 14, 2021, the first year of the pandemics. The data were compared with the same period of the previous year.
The authors compiled several data and analyzed them considering three different moments, named “lockdown”, “unlock” and “second wave”. A series of statistical analyzes were performed to allow a proper interpretation of the data. The authors show evidence of the impact of COVID-19 in decreasing surgical volume compared to the pre-pandemic period, and discuss how this may impact human health and what the medical community should consider in the event of a future health crisis.
The study was well designed and the results are solid with good discussion. However, this manuscript is too specific to be published in a journal like Applied Science and I believe it would fit better in a journal focused on ophthalmology.
Response 1: We greatly appreciate your comments and suggestions. We received a kind invitation from the Editor of the Special Issue to contribute with a paper focused on Ophthalmology practice and thus we proceeded accordingly.
Point 2: That said, I suggest the authors improve the quality of figure 3. The resolution is very low, which makes it difficult to read the numbers in the graphs. Also, on page 12 the authors mention table 10, but it is not shown. I believe they refer to table 8. If so, this should be corrected. Finally, there are some typos throughout the text that should be corrected. A careful check will resolve the issue.
Response 2: Thank you for your comment. Correction has been made in the revised manuscript.
Round 2
Reviewer 1 Report
Dear authors, please find below indications about your manuscript:
Statistical analysis:
- Please consider that the linear regression analysis rests on the assumption that the dependent variable is continuous
- Table 8: table is not clear and the results are confusing, R2 is not adequate to explain the relationship between outcome and covariates in a linear regression model
Author Response
Point 1: Please consider that the linear regression analysis rests on the assumption that the dependent variable is continuous
Response 1: Thank you for your comment, we have reviewed the sentence from line 187 to 190
Point 2: Table 8: table is not clear and the results are confusing, R2 is not adequate to explain the relationship between outcome and covariates in a linear regression model
Response 2: we have modified the table making it more understandable and removing R2
Reviewer 2 Report
Thank you for the amendments. The presentation is clearer to me now.
Line 39: unknown cases of pneumonia - does it mean pneumonia of unknown cause?
Author Response
Point 1 : Line 39: unknown cases of pneumonia - does it mean pneumonia of unknown cause?
Response 1: thank you for your comment, correction has been made in the revised manuscript.
Reviewer 3 Report
The authors fully addressed all points I've raised and improved the manuscript accordingly. It is now suitable for publication. Congratulations to the authors for the good work!
Author Response
thanks for your comment
Round 3
Reviewer 1 Report
Dear authors,
methodological issues persist on the data analysis, especially on the type of regression (not suitable for the outcome under study) and on the results (R? Maybe the authors mean a correlation analysis, but this aspect is not clear from the manuscript). I suggest a support for the statistical analysis.
Author Response
Point 1 methodological issues persist on the data analysis, especially on the type of regression (not suitable for the outcome under study) and on the results (R? Maybe the authors mean a correlation analysis, but this aspect is not clear from the manuscript). I suggest a support for the statistical analysis.
Response 1: Thank you for pointing out, we followed your suggestion and with the support of a statistician we made the decision to remove the inferential statistic on the relationship between COVID-19 patients and surgical procedures. Therefore, we just described the different trend of COVID-19 patients hospitalised and in ICU in both regions in the result section, and added a box to Figure 1 summarizing the overall change in surgical procedures for each study period